# Event-Assisted Object Tracking on High-Speed Drones in Harsh Illumination Environment

Yuqi Han [1], Xiaohang Yu [2], Heng Luan [3] and Jinli Suo [1,*]

1   Department of Automation, Tsinghua University, Beijing 100084, China; yqhan@mail.tsinghua.edu.cn
2   Tsinghua-UC Berkeley Shenzhen Institute, Shenzhen 518071, China; yuxh21@mails.tsinghua.edu.cn
3   Research and Development Center, TravelSky Technology Ltd., Beijing 101318, China; luanheng@travelsky.com.cn
*   Correspondence: jlsuo@tsinghua.edu.cn

**Abstract:** Drones have been used in a variety of scenarios, such as atmospheric monitoring, fire rescue, agricultural irrigation, etc., in which accurate environmental perception is of crucial importance for both decision making and control. Among drone sensors, the RGB camera is indispensable for capturing rich visual information for vehicle navigation but encounters a grand challenge in high-dynamic-range scenes, which frequently occur in real applications. Specifically, the recorded frames suffer from underexposure and overexposure simultaneously and degenerate the successive vision tasks. To solve the problem, we take object tracking as an example and leverage the superior response of event cameras over a large intensity range to propose an event-assisted object tracking algorithm that can achieve reliable tracking under large intensity variations. Specifically, we propose to pursue feature matching from dense event signals and, based on this, to (i) design a U-Net-based image enhancement algorithm to balance RGB intensity with the help of neighboring frames in the time domain and then (ii) construct a dual-input tracking model to track the moving objects from intensity-balanced RGB video and event sequences. The proposed approach is comprehensively validated in both simulation and real experiments.

**Keywords:** drones; harsh illumination; image enhancement; event-assisted object tracking; multi-sensor fusion





## 1. Introduction

As lightweight, flexible, and cost-effective [1–3] platforms, drones have often been used in a variety of remote tasks, such as surveillance [4,5], detection [6], and delivery [7]. In such applications, drones need to accurately perceive the surrounding environments to support subsequent decisions and actions. In general, common sensors used on UAVs include visible-wavelength optical cameras [8], LiDAR [9], NIR/MIR cameras [10], etc. Each type of sensor has its own advantages and disadvantages, so multi-mode sensing has been the typical solution in this field. Among the various sensors, the visible-wavelength camera is an indispensable sensing unit due to its high resolution, capability of collecting rich information, and low cost of construction.

As one of the most important tasks of a drone, object tracking [11–14] has been widely studied. Broadly speaking, object-tracking algorithms take either the RGB frame as input or its combination with other sensing modes. RGB-only methods [15–18] prevail in frame-based object tracking but are limited in harsh illumination scenarios. Some researchers proposed to incorporate information from event-based cameras, which show superior performance in both low-light and high-dynamic-range scenes. To fuse the information from RGB frames and event sequences, Mitrokin et al. [19] proposed a time-image representation to combine temporal information of the event stream, and Chen et al. [20] improved event representation by proposing a synchronous Time-Surface with Linear Time Decay

representation. These approaches exhibit promising performance in object tracking with high time consistency.

However, the above methods are difficult to apply on 24/7 UAVs due to the limited sensing capability of RGB sensors in cases with complex illumination. Because overexposure and underexposure both lead to the image quality degrading greatly and hamper accurate tracking, the reliable drone-based sensing of harshly lit scenes is quite challenging [21,22]. Taking the video in Figure 1 as an example, when capturing a car traveling through a tunnel, there exist a large intensity range in each frame and abrupt variation among different frames; the car is even undetectable in some frames by both tracking algorithms and human vision systems due to underexposure. Fortunately, drones are subjected to continuously varying illumination while in flight, causing recordings with different quality for a target region. Considering that the feature points of neighboring video frames are mostly consistent [23,24], we are inspired to compensate low-quality images with guidance from high-quality counterparts, achieving continuously high-quality videos as well as robust downstream tasks. One of the most crucial problems is to recognize the matching features in adjacent frames that undergo abrupt intensity changes.

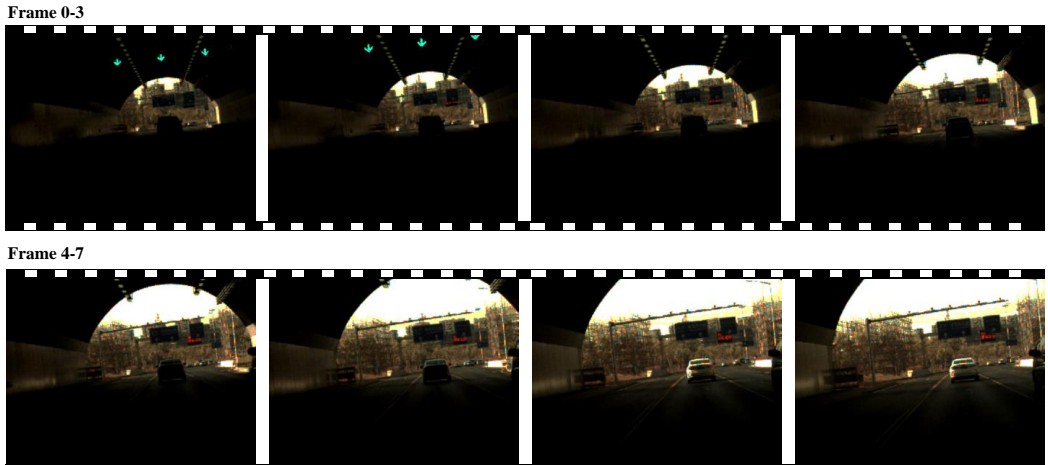

**Figure 1.** A typical high-dynamic-range RGB video of a car driving through a tunnel, in which the car is almost invisible in the last 5~6 frames due to underexposure.

To increase the image quality under harsh illumination, researchers have made a lot of explorations in recent years. One most common way is to reconstruct HDR images by merging the set of multi-exposure LDR images [25]. For dynamic scenes, image alignment is required to address the inconsistency among frames with different exposures. Kang et al. [26] initially aligned neighboring frames with the reference frame and merged these aligned images to craft an HDR image. Later works [27,28] modified it by adding a motion estimation block and a refinement stage. Differently, Kalantari et al. [29] proposed a patch-based optimization technique, synthesizing absent exposures within each image before reconstructing the ultimate HDR image. Gryaditskaya et al. [30] enhanced this method by introducing an adaptive metering algorithm capable of adjusting exposures, thereby mitigating artifacts induced by motion. Instead of capturing frames with different exposure times, some methods use deep neural networks to reconstruct the HDR image from a single input image. However, due to relying on a fixed reference exposure, the reconstruction is strongly ill-posed and cannot achieve high between-frame consistency. Additionally, many existing HDR video reconstruction methods focus on developing some special hardware, such as scanline exposure/ISO [31–33], per-pixel exposure [34], modulo camera [35], etc., but these new cameras are still being research and not ready for commercial use in a near future. Some other recent approaches work under the deep-optics scheme and focus on jointly optimizing both the optical encoder and CNN-based decoder for HDR imaging challenges. The above methods usually make assumptions about the lighting

conditions, which might not hold in real scenes. Additionally, most of these algorithms need ground-truth high-dynamic-range images for supervised network training and exhibit limited performance in scenes different from the training data. Hence, these methods are enlightening but difficult to be directly applied on practical UAV platforms working in open environments.

The event camera, also known as neuromorphic vision sensor, is an emerging technique that records intensity changes exceeding the threshold asynchronously [36,37]. In recent years, event signals have been used in a variety of high-speed tasks due to their high sensitivity and fast response, such as high-speed tracking [38–41], frame interpolation [42,43], optical flow estimation [44–46], motion detection [47], etc. Unlike conventional optical camera sensors, event cameras output the "events" indicating that there occurs sufficiently large intensity variation at certain positions and instants and also indicate the polarity of the change. Considering that an event camera can record the motion over a large intensity range and is insensitive to abrupt intensity changes, we propose to use event signals to explicitly align the RGB frames and thus compensate for the quality degradation harming the successive object tracking. In other words, with the consistent description of event signals, we enhance low-quality images under guidance from their high-quality counterparts and achieve continuous high-quality scene perception. Specifically, we match the key points occurring at different instants [48] and utilize the matching to balance the intensity change in sequential RGB frames. Afterward, we construct a fusion network to aggregate the enhanced RGB frames and event signals for robust object tracking.

The contributions of this paper are as follows:

- We propose an event-assisted robust object-tracking algorithm working in high-dynamic-range scenes, which successfully integrates the information from an event camera and an RGB camera to overcome the negative impact of harsh illumination on tracking performance. As far as we know, this is the first work of object tracking under harsh illumination using dual-mode cameras.
- We construct an end-to-end deep neural network to enhance the high-dynamic-range RGB frames and conduct object tracking sequentially, and the model is built in an unsupervised manner. According to the quantitative experiment, the proposed solution improves tracking accuracy by up to 39.3%.
- We design an approach to match the feature points occurring at different time instants from the dense event sequence, which guides the intensity compensation in high-dynamic-range RGB frames. The proposed feature alignment can register the key points in high-dynamic-range frames occurring within a 1 s window.
- The approach demonstrates superb performance in a variety of harshly lit environments, which validates the effectiveness of the proposed approach and largely broadens the practical applications of drones.

In the following, we first introduce the framework and algorithm design for the proposed event-assisted object tracking in Section 2, including event-based cross-frame alignment, RGB image enhancement, and dual-mode object tracking. In Section 3, we present the experimental settings, including the datasets and training details. The qualitative results and quantitative results are discussed in Section 4. Further, we present the results for the real-world data as well as the ablation study. Finally, in Section 5, we summarize the paper, discuss the limitation of the proposed solution, and highlight future work to be conducted on efficient collaborative sensing around drones.

## 2. Framework and Algorithm Design

This section presents the details of the proposed event-assisted robust object-tracking approach working under harsh illumination. Here, we first briefly introduce the framework and then describe the design of three key modules, including the retrieval of feature registration across frames, the enhancement of high-dynamic-range frames, and the successive dual-mode object tracking.

The basic idea of the proposed approach is to utilize the reliable motion cue perception capability of event cameras to prevent the quality degradation of RGB frames and then combine the event signals and the enhanced RGB video to boost the successive tracking performance suffering from overexposure and underexposure. The whole framework of the proposed event-assisted object-tracking approach is shown in Figure 2; it consists of mainly three key modules:

(i)　Retrieving the motion trajectories of key feature points from the dense event sequence. We divide the event sequence into groups occurring in overlapping, short time windows, and the key points from Harris corner detection in each event group can construct some motion trajectories. Further, we integrate these short local trajectories to figure out the motion over a longer period across the RGB frames, even under harsh illumination.

(ii)　Enhancing the high-dynamic-range RGB frame according to inter-frame matching and information propagation. Based on the matching among feature points across frames, we build a deep neural network to compensate for the overexposed or underexposed regions using neighboring frames with higher-visibility reference frames to guide low-visibility objective frames. In implementation, we build a U-Net-based neural network for image enhancement.

(iii)　Tracking the target objects by fusing information from both RGB and event inputs. We design a tracking model taking dual-mode inputs to aggregate the information from the enhanced RGB frames and event sequences to locate the motion trajectories. Specifically, we construct 3D CNNs for feature extraction, fuse the features from two arms using the self-attention mechanism, and then employ an MLP to infer the final object motion.

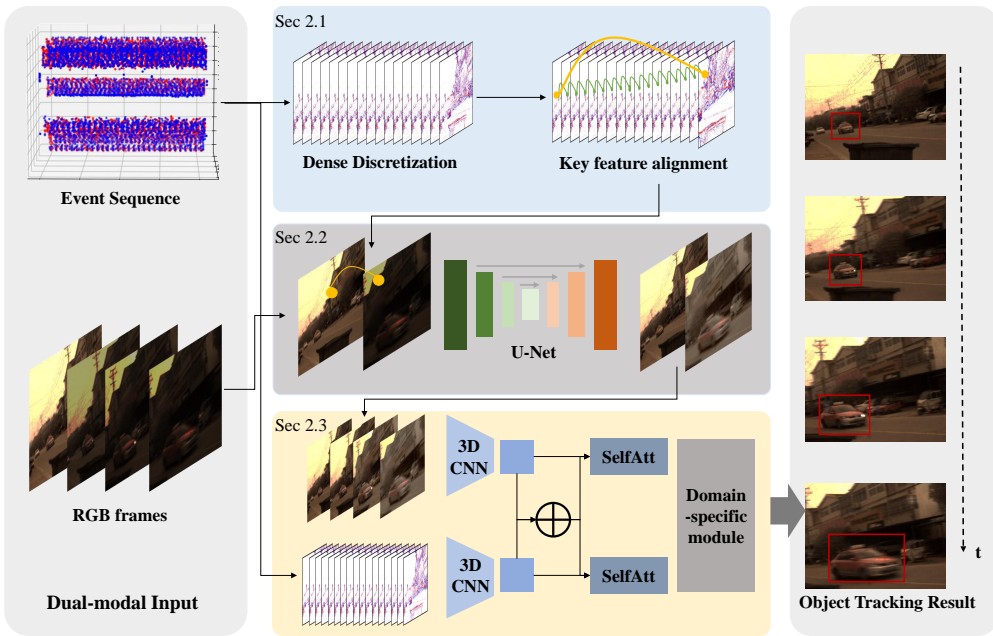

**Figure 2.** The framework and working flow of the event-assisted robust tracking algorithm under harsh illumination. The whole pipeline is fully automatic and consists of three key steps, with the first one including conventional optimization and the latter two being implemented by deep neural networks.

## 2.1. Event-Based Cross-Frame Alignment

Event-based key feature extraction and matching are conducted here to utilize the stable event signals under harsh illumination for cross-frame alignment of the degraded RGB video, facilitating frame compensation using corresponding positions with decent

quality in neighboring frames. We locate the key features of moving objects from the event sequence, as illustrated in Figure 3.

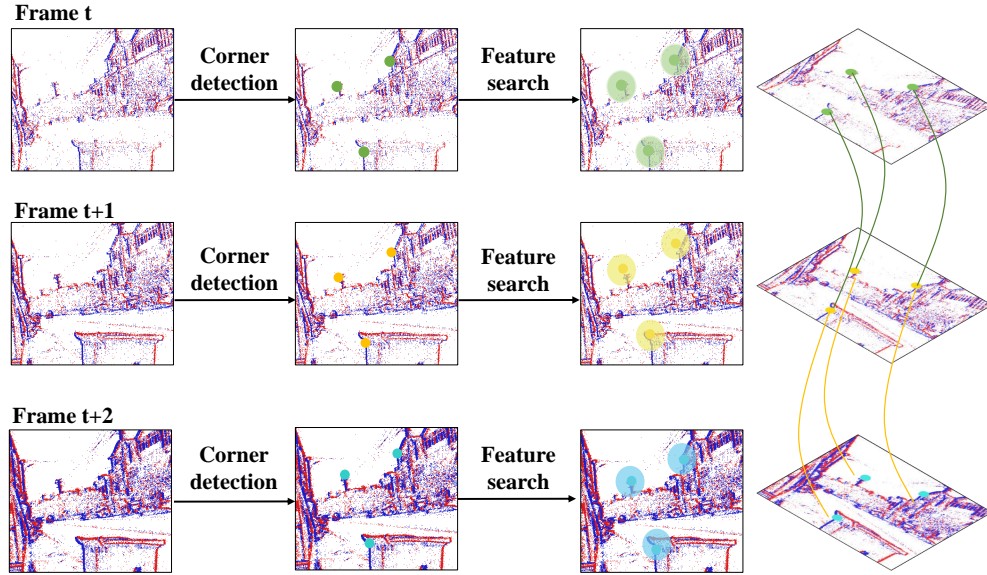

**Figure 3.** The illustration of event-based key point alignment. We locate the key feature points through Harris detection and search the matching counterparts locally (circular candidate regions are highlighted with different colors) to constitute the motion trajectories, as shown in the right column.

Given a time duration $T$, we assume that there are $N$ RGB frames and $S$ event signals. We define the captured RGB frames as $\{I_0, I_1, \cdots, I_N\}$, and the corresponding time stamps are defined as $\{T_0, T_1, \cdots, T_N\}$. Event signal $s$ is defined as a quadruple $x_s, y_s, t_s, P_s$, where $x_s, y_s$ denote the coordinates of $s$; $t_s$ presents the response time instant; and $P_s$ indicates the polarity of intensity change. Firstly, we divide the $S$ event signals into $K \times N$ groups along the time dimension and project each group into $K \times N$ 2D images, named event frame. We adopt the Harris corner detection algorithm for the above event frames to extract individual key feature points. Further, we align the key feature points at different time instants. Assuming that the shape of the moving objects is fixed within a very short time slot, i.e., the key features in adjacent frames are similar, we construct a small circular search region with radius $r$ around each key feature. In other words, the key feature at the $e$th frame matches the features inside the searching circle of the $e + 1$th frame.

For the $n$th RGB frame, we first align the event frames between $n \times S$ and $(n + 1) \times S$. From the displacement between the features of multiple event frames, one can construct the moving trajectory of the key event feature points, which reflects the displacement of the corresponding key features in the RGB frame. Naturally, we can eventually infer the position of the corresponding key feature from $n$th to $n + 1$th RGB frames.

### 2.2. RGB Image Enhancement

After matching the feature points in different RGB frames, we enhance the underexposed and overexposed regions utilizing the high-visibility counterparts to adjust the intensity and supplement the details. For description simplicity, we define the low-visibility frames as the objective and the high-visibility frames as the reference. To achieve enhancement, there are two core issues to be addressed: (i) how to determine the objective frame that needs to be enhanced; (ii) how to design the learning model to improve the visibility to match the reference frame while preserving the original structure of the objective frame.

We first estimate the visibility of the frames to determine which frames are highly degraded. Intuitively, since harsh illumination leads to local overexposure or underexposure, which is usually of lacking texture, we use information richness to characterize the

degeneration degree. In implementation, we define the visibility ($V_i$) of input RGB image $R_i$ as the difference from its low-pass-filtered version ($\hat{R}_i$), i.e.,

$$V_i = Var(R_i - \hat{R}_i), \tag{1}$$

where $Var(\cdot)$ denotes the variance calculation.

In general, we divide the frames into groups and conduct compensation within each group. We iteratively find the objective frame with the lowest visibility score and the reference frame with the highest visibility and then conduct enhancement. The iteration ends when the number of iterations exceeds a predetermined number $P$ or the difference between the visibility of the target and the reference frame smaller than $\eta$. In our experiments, we set $P = 10$ and $\eta = 0.1$.

For enhancement, we designed a U-Net-shaped network structure inferring the enhanced frame from the objective and reference frames, as shown in Figure 4. (The corresponding optimization process is detailed in [49].) The network consists of a three-layer encoder for feature extraction and a three-layer decoder. Skip connections are used to facilitate the preservation of spatial information. The network is trained in an unsupervised manner. We define the loss function based on aligned feature points. Considering that the enhanced frame is expected to be similar to the reference image around the key feature points and close to the original frame at other locations, we define a combinational loss function. To guarantee the former similarity, we minimize the MSE difference, and we use the LPIPS loss for the latter. Denoting the reference image by $I_\text{ref}$, the original objective image as $I_\text{obj}$, and the output as $I_\text{out}$, we define the loss function as

$$L = MSE(I_\text{ref}(k) - I_\text{out}(k)) + \alpha LPIPS(I_\text{obj}(\neg k) - I_\text{out}(\neg k)) \tag{2}$$

where $k$ denotes the positions of key features and $\neg k$ denotes the remaining pixels; $\alpha$ is the hyper-parameter, which is set to 0.05 during training.

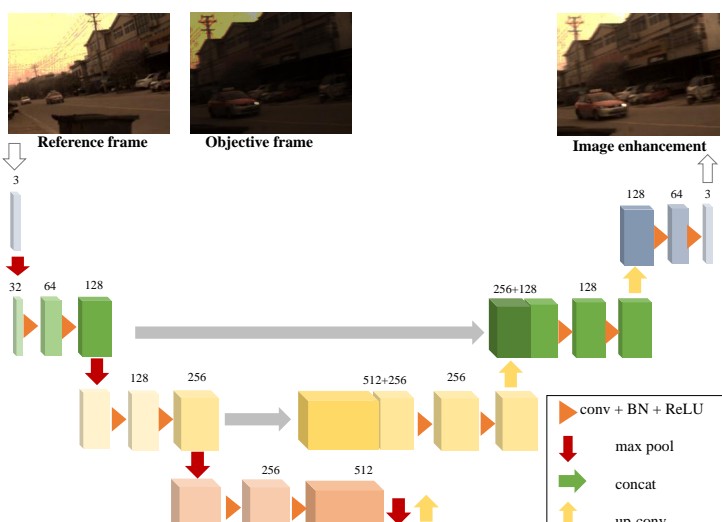

**Figure 4.** The structure of the RGB image enhancement module. We input the captured RGB frame into the U-Net network, which comprises a three-layer encoder for feature extraction and a three-layer decoder for image enhancement. The network includes skip connections to connect encoder and decoder layers, facilitating the preservation of spatial information. This diagram showcases convolutional, pooling, upsampling, and downsampling layers, with the following key operations: conv denotes convolution; BN denotes batch normalization; ReLU refers to the ReLU activation function; max pool denotes the max pooling operation; concat and Up-conv denote the concatenation and transposed convolution, respectively.

*2.3. Dual-Mode Object Tracking*

To leverage the motion cues in both the event sequence and the enhanced RGB frames, we construct a dual-mode tracking module for reliable object tracking. The proposed dual-mode tracking module is based on RT-MDNet [50]. The module consists of a shared feature mapping network aiming at constructing the shared representation to distinguish the object from the background and a domain-specific network focusing on domain-independent information extraction. Different from RT-MDNet [50], the proposed dual-mode design focuses on feature fusion from two types of inputs and constructs two self-attention modules to highlight the combinational representation from two individual inputs.

The architecture of the network is shown in Figure 5. We first construct two individual 3D CNNs to extract features from the inputs and output feature vectors of the same size. Subsequently, we concatenate the two feature vectors and use convolution to obtain a combinational representation of the fused features. Subsequently, we construct the self-attention network to retrieve the information underlying independent feature inputs. (Please refer to [51] for the steps of the optimization process.) A two-layer fully connected MLP is used to output the common feature. We refer to RT-MDNet [50] to construct the domain-specific layer afterward, outputting the final tracking results.

During model training, for each detection bounding box, a cross-entropy loss function is constructed to ensure that the target and background are separated as much as possible, and the same also applies to multiple domains. In the latter, fine-tuning stage, we apply different strategies for the first frame and the subsequent ones of a given sequence. For the first frame, we choose multiple bounding boxes following a Gaussian distribution to conduct domain-specific adaption, while for the subsequent frames, we build random samples based on the results from the previous frame and search for the proper bounding box through regression.

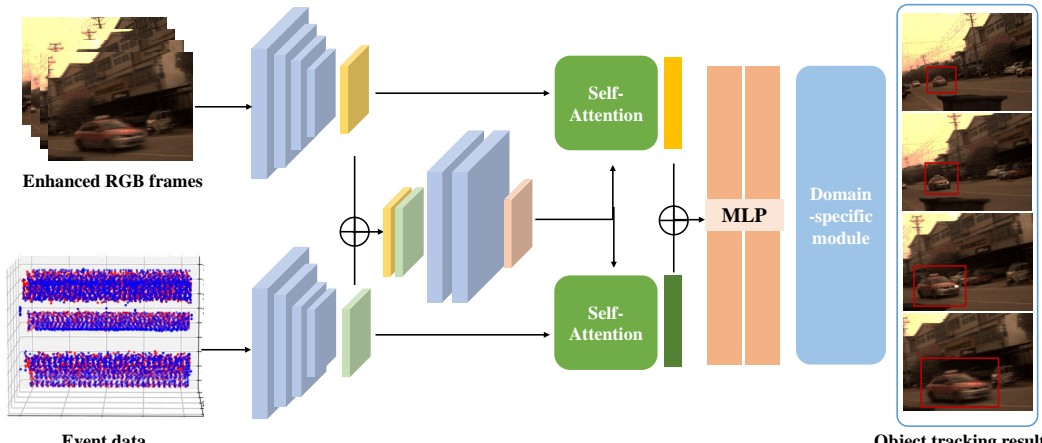

**Figure 5.** The structure of the object-tracking module. The RGB frames and event sequence are individually fed into two 3D CNN modules for feature extraction, and the extracted features are concatenated and sent to another CNN module for fusion. Then, the individual and fused features are separately sent to the self-attention network. Finally, two MLPs are applied to derive the object detection and tracking results.

## 3. Experimental Settings

**Datasets.** We verify the proposed method on both simulated and real datasets. We use VisEvent [52] as the simulated data and mimic harsh illumination by modifying the brightness and contrast of the RGB frames. Specifically, we modify the luminance and contrast as follows: We let the luminance vary linearly, quadratically, or exponentially across the frames, and the image contrast undergoes a linear change with different slopes. We first randomly select 1/3 of the data for luminance modification and then apply contrast modification to 1/3 randomly selected videos. Two examples from the simulated dataset

are shown in Figure 6. The first scene mimics the brightness changes in the underexposed scenes, and the second scene simulates overexposure, through modification of image brightness and contrast. One can see that we can generate videos under complex illumination from the original counterpart with roughly uniform illuminance. In the generated high-dynamic-range RGB frames, the textures of some regions are invisible in some frames due to either underexposure or overexposure. In contrast, the contours across the whole field of view are recorded decently.

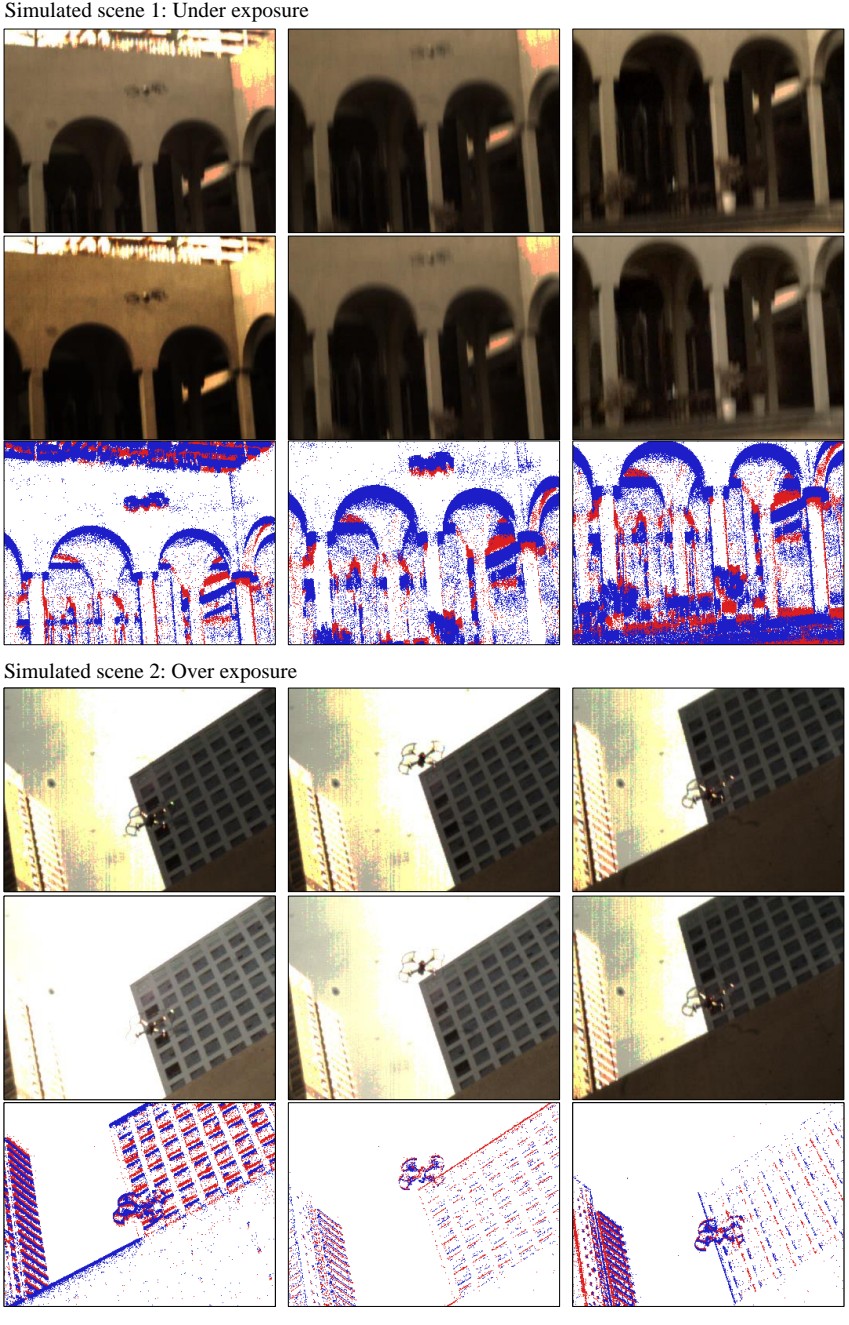

**Figure 6.** Two exemplar scenes from the simulated high-dynamic-range videos based on the VisEvent dataset. For each scene, we list the original RGB frames, the synthetic high-dynamic-range frames, and the corresponding events from top to bottom. The first scene has a linear increase in intensity and a linear decrease in contrast to mimic underexposure in the 1st frame. The second sequence undergoes linearly decreasing intensity to mimic overexposure in the first frame.

For the real-world data, we captured some typical nighttime traffic scenes with a pair of registered cameras (one RGB and the other events). The scenes consist of complex illumination (e.g., traffic lights, neon signs, etc.) and large intensity variations. From the two exemplar scenes in Figure 7, it can be seen that these scenarios exhibit large illuminance variations and the traffic participants are almost invisible in some frames, due to either underexposure or overexposure. This challenging dataset can be directly used to test the effectiveness of the proposed object-tracking algorithm in real scenarios, as shown in Figure 7.

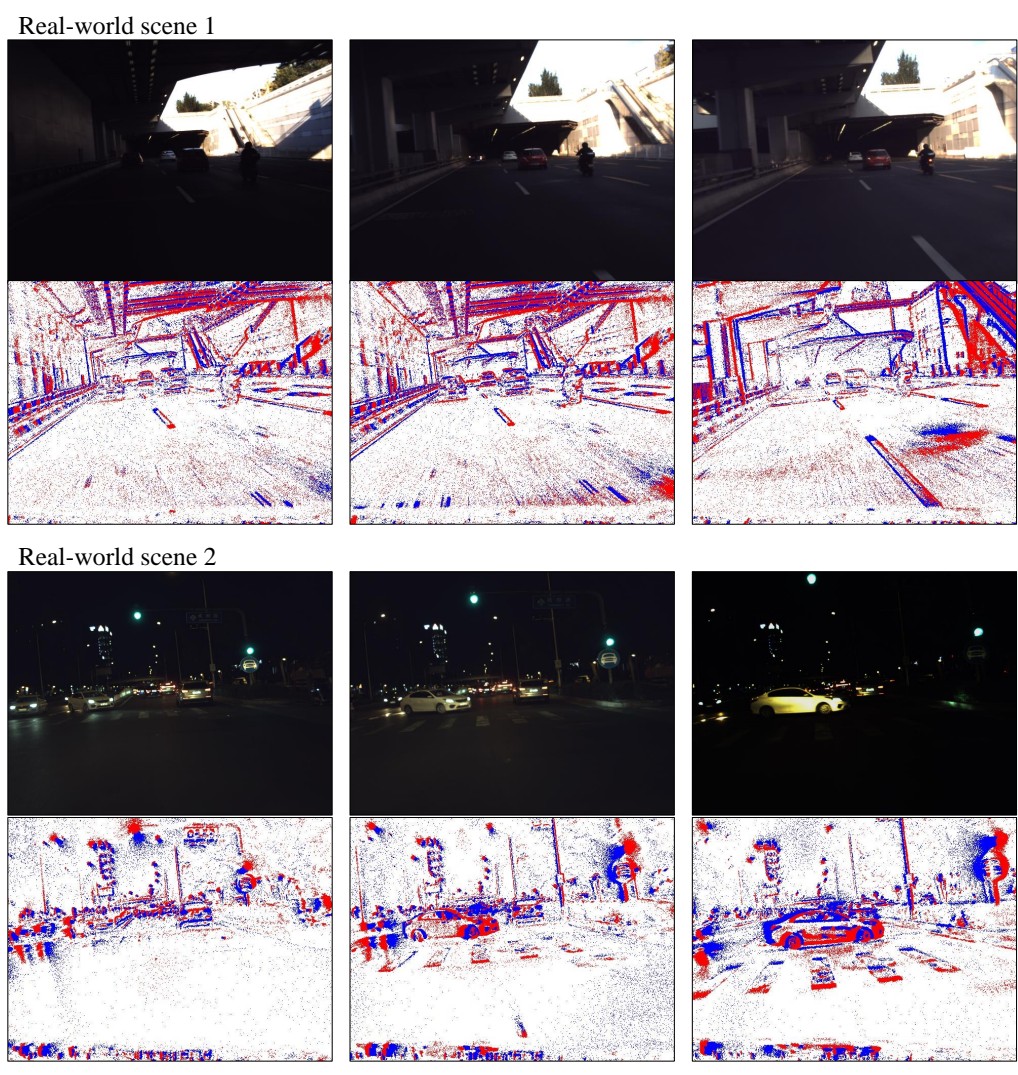

**Figure 7.** Two typical examples from the real-world dataset captured in harshly lit traffic scenarios, collected under a bridge during the daytime and at a crossroad at night, respectively. For each scene, the RGB and event cameras are pre-calibrated for pixel-wise registration.

**Baseline algorithms.** We choose three different algorithms with state-of-the-art tracking performance as baselines for the proposed solution, i.e., RT-MDNet [50], Siamrpn++ [53], and VisEvent [52]. RT-MDNet [50] and Siamrpn++ [53] are two RGB-input trackers performing well under normal illumination. So far, there are few objective algorithms specially developed for harsh illumination scenarios; we chose the above two robust and widely used tracking solutions as baselines. VisEvent [52] constructs a two-modality neural network fusing RGB and event signals, and we compare the proposed solution with VisEvent [52] to verify the effectiveness of the image enhancement module under harsh illumination. This benchmark has input similar to our method's and exhibits state-of-the-art performance,

serving as a good option to validate the proposed image enhancement module under harsh illumination.

**Training.** Training is implemented on the NVIDIA 3090 for about 4.7 h. We set the input image size as well as the spatial resolution of the event sequence to $640 \times 480$ pixels and seven continuous RGB frames ($\sim$350 ms) for intensity balancing. We use the Adam optimizer, with the learning rate being $5 \times 10^{-4}$, the momentum being 0.9, and the weight decay being $5 \times 10^{-4}$.

## 4. Results

In this section, we construct a series of experiments to verify the effectiveness of the proposed method on two tasks in high-dynamic-range scenes: image enhancement and object tracking. We first show the visual and quantitative performance against some baseline algorithms. Also, we give the qualitative results based on the real data to further show the visual difference between the proposed solution and baselines. Finally, we conduct ablation experiments to quantify the contribution of the key module of the algorithms.

### 4.1. Results Based on Simulated Data

In this subsection, we validate our approach in terms of image enhancement and object-tracking accuracy, based on simulated data. Here, we give both qualitative and quantitative experimental results to comprehensively analyze the effectiveness of the proposed solution. For the qualitative results, we show the result of image enhancement first and compare the object-tracking performance with that of the baseline algorithms afterwards. For the quantitative results, we compare the precision plot (PP) and success plot (SP) to assess the tracking performance.

#### 4.1.1. Qualitative Results

Figure 8 shows the qualitative results for an exemplar video from the simulated dataset. The top row shows the raw RGB sequence, with large intensity changes both within and across frames. In this scene, a person runs from a location with strong illumination toward a destination with a large shadow. Due to the extremely dark intensity, it is challenging to recognize their silhouette in the last frame. We enhance the RGB frames according to the temporal matching extracted from the event signals, and the results are shown in the middle row. The enhanced version is of much more balanced intensity and can highlight the human profile even under weak illumination.

We further show the object-tracking result in the bottom row. The bounding boxes of our approach and the other three competitors are overlaid, with different colors. When sufficiently illuminated, all the algorithms can track the object with high accuracy. RT-MDNet, VisEvent, and the proposed algorithm are comparable, while there exists some deviation in the bounding box output by Siamrpn++ tracking. When the light becomes weak, the proposed algorithm can still identify the person's location, while RT-MDNet's and VisEvent's bounding boxes deviate. When the light is extremely weak, only the proposed method, RT-MDNet, and VisEvent can track the object, because of high sensitivity and robustness to abrupt intensity changes in the event signals. In comparison, the RGB image in RT-MDNet and VisEvent is not enhanced and thus reduces the final tracking accuracy, while our approach demonstrates reliable tracking consistently.

#### 4.1.2. Quantitative Results

We introduce the typical matrix PP and SP here to evaluate accuracy in object tracking. Specifically, the PP indicates the frame percentage where the deviation between the estimated object center location and ground truth is less than the determined threshold. The SP denotes the frame percentage where the IoU between the estimated bounding box and the ground-truth bounding boxes is higher than the determined threshold. Table 1 shows the PPs and SPs of our approach and three state-of-the-art object-tracking algorithms.

Since there is no ground truth for the real data, we only conduct quantitative analysis on simulated data.

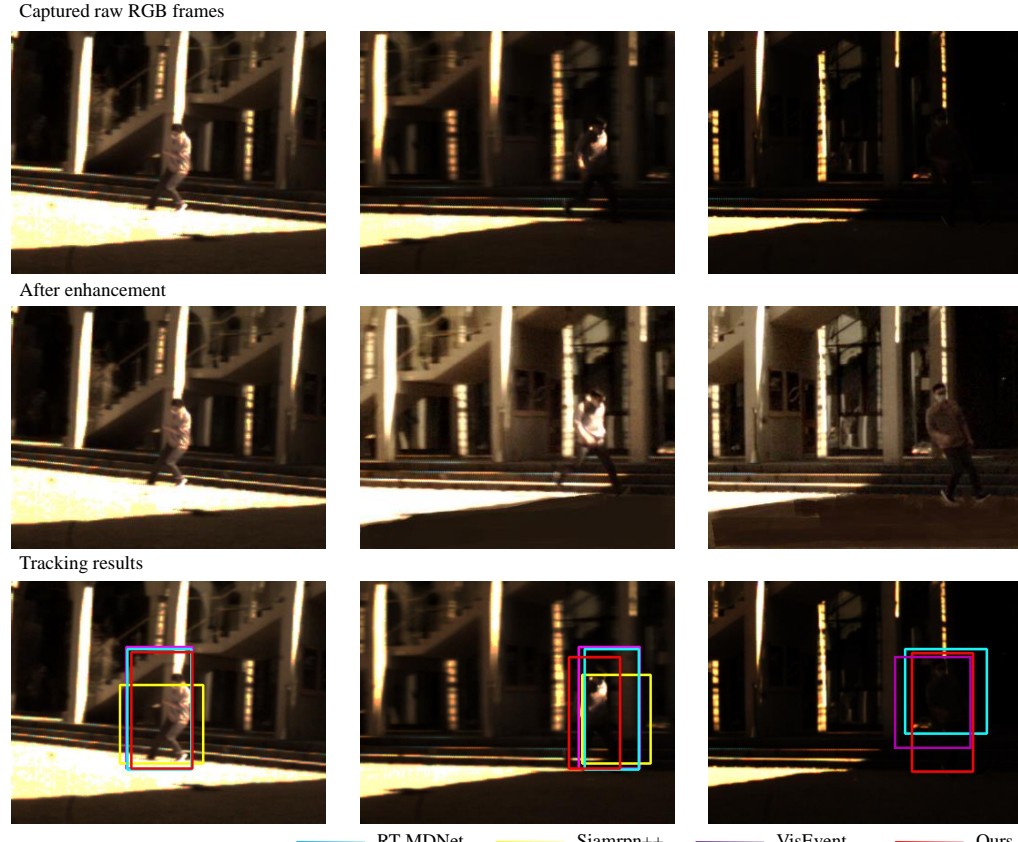

**Figure 8.** Performance of RGB enhancement in object tracking in a typical exemplar scene in the simulated dataset. (**Top**) The captured RGB video frames. (**Middle**) The corresponding enhanced images obtained with the proposed method. (**Bottom**) The tracking results of different object-tracking algorithms.

**Table 1.** The quantitative performance of different object-tracking algorithms on the simulated dataset, in terms of PPs and SPs.

|  | **Our Algorithm** | **VisEvent** | **Siamrpn++** | **RT-MDNet** |
|---|---|---|---|---|
| PP | **0.783** | 0.712 | 0.390 | 0.405 |
| SP | **0.554** | 0.465 | 0.232 | 0.321 |

According to Table 1, the proposed algorithm demonstrates the tracking results with the highest accuracy. Even under harsh illumination, we can track the target object continuously, while Siamrpn++ and RT-MDNet show poor tracking results under the same conditions. Moreover, though VisEvent takes the event signal as the input, it ignores the influence of the low-quality RGB frames and produces inferior tracking accuracy. From the ranking, we can draw two conclusions: first, the event signals can help address performance degeneration in high-dynamic-range scenes; secondly, enhancing the degraded RGB frames can further raise accuracy in object tracking.

*4.2. Results Based on Real-World Data*

To investigate the performance of our approach in real high-dynamic-range scenes, we test our algorithm on some videos under challenging illumination, with one typical example being shown in Figure 9. The video is captured at a tunnel entrance, and the

frames in the top row show a car traveling through the tunnel. When the car enters the tunnel, it is difficult to capture images with high visual quality due to insufficient light, and the car turns indistinguishable in the last frame. The middle row shows the result of image enhancement, demonstrating that the visual quality of the RGB frames is largely increased compared with the raw input.

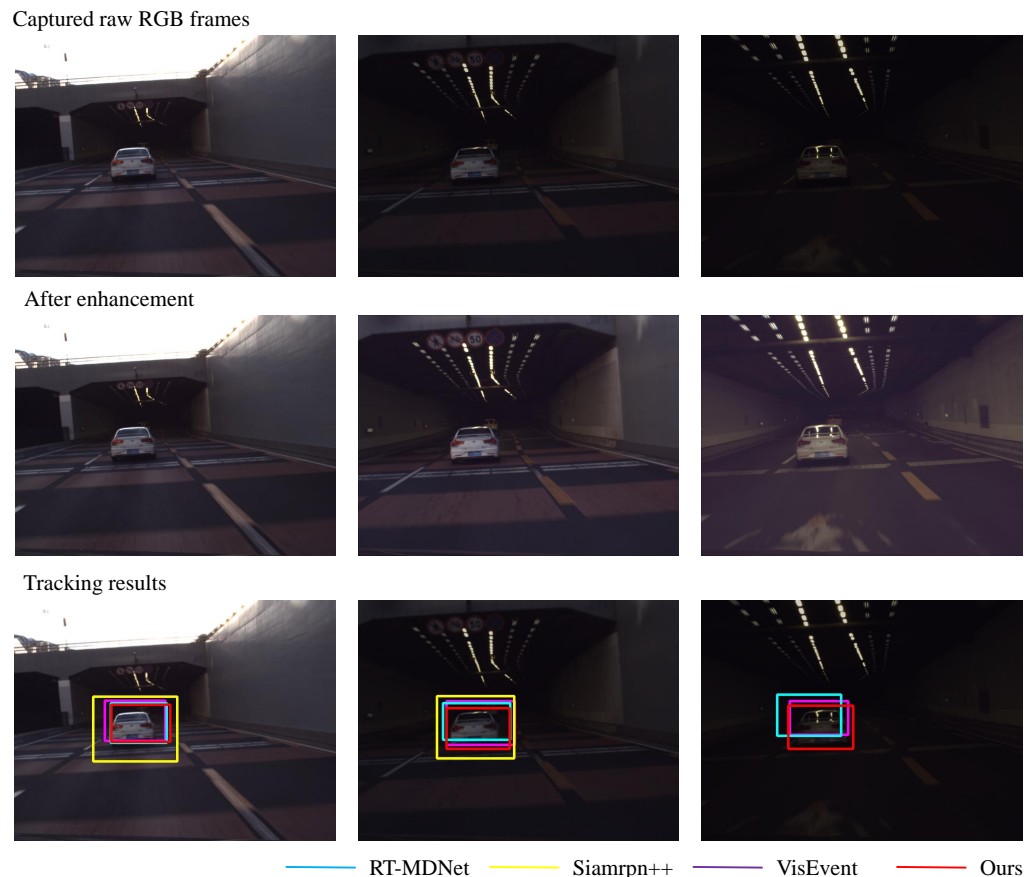

**Figure 9.** Demonstration of our image enhancement of the tracking result and performance comparison with existing object-tracking algorithms on a real-world high-dynamic-range scene—a white car driving through a tunnel. (**Top**) The captured RGB frames. (**Middle**) Our enhanced RGB images. (**Bottom**) The tracking results of different algorithms.

The tracking results are shown in the bottom row of Figure 9. All four algorithms can track the car at high brightness. When the light becomes weaker, the performance of the two RGB-based tracking algorithms decreases: Siamrpn++ cannot track the car, and RT-MDNet produces a bounding box with a large offset; on the other hand, VisEvent can achieve relatively higher robustness, but the bounding box is not accurate. On the contrary, we can achieve reliable tracking over the whole sequence. Based on the above experiments, we can further verify that (i) the illumination condition affects accuracy in object tracking and (ii) the event signal can assist object tracking under harsh illumination.

### 4.3. Ablation Studies

The ablation experiment focuses on validating the contribution of event-based temporal alignment to RGB image enhancement and object tracking. In the proposed approach, we use Harris corner detection to retrieve key feature points from the dense event sequence, and here, we compare its performance against two methods: using random event signals as key features and using the detected Harris corner points from the RGB images rather than event signals.

From the upper row in Figure 10, one can see that there exist large intensity variations within each frame and abrupt changes among frames, which is quite challenging for object-tracking algorithms and even human vision systems, especially in the third frame. Here, we adopt the person in the third frame as the tracking target, and the results with different key feature guidance are shown in the bottom row. One can see that the proposed alignment strategy performs best in terms of both the quality of the enhanced image and object-tracking accuracy. In comparison, the result produced through registration from random event signals slightly enhances image quality and results in a looser bounding box, while registration from RGB frames provides little help, which again validates the strategy of introducing event cameras for such harshly lit scenes. The inferior performance of the two benchmarking implementations is mainly attributed to the fact that they cannot identify the temporal matching properly due to the lack of descriptive features.

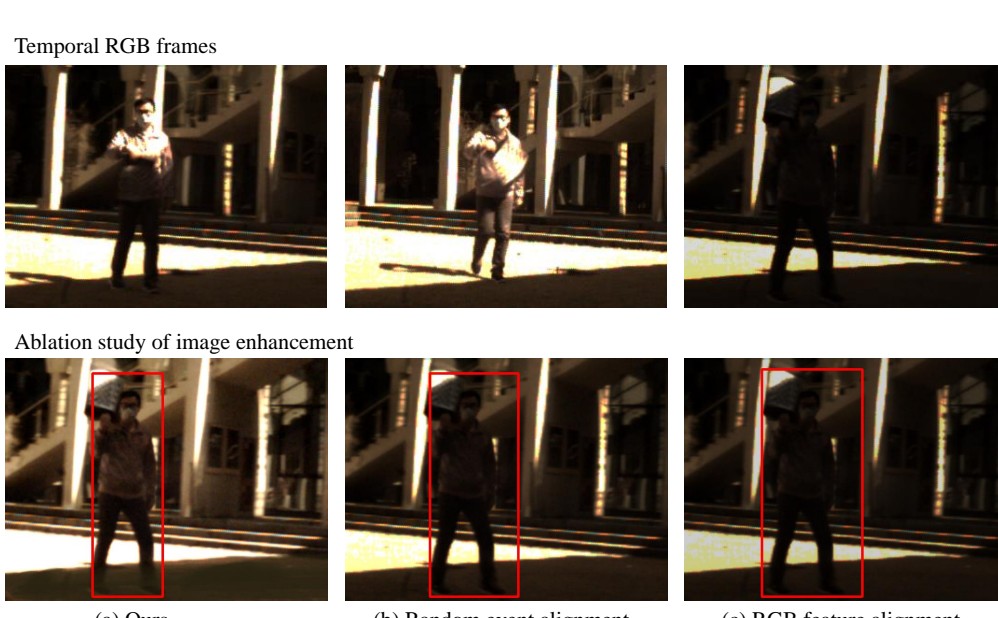

**Figure 10.** An example showing the results of ablation studies. The upper row displays the RGB frames of a high-dynamic-range scene. The lower row shows the image enhancement and object-tracking results based on three different types of temporal registration guidance, with the person in the third frame (the darkest and most challenging one) as the target object. From left to right: key feature alignment using the proposed event-based Harris corner points, random event signals, and Harris corner points in RGB frames.

## 5. Summary and Discussions

Visible-wavelength optical cameras provide rich scene information for the environmental sensing of drones. However, harsh illumination causes high dynamic ranges (e.g., at nighttime, at entrances or exits, etc.) and hampers reliable environmental perception. In order to extend the applicability of visible-wavelength cameras in real scenes, we propose a dual-sensing architecture that leverages the advantages of event cameras to increase the imaging quality of the RGB sensor as well as the successive object-tracking performance.

The proposed event-assisted robust object tracker exploits two main features of event signals, i.e., robust imaging under complex illumination and fast response. These advantageous and unique features support extracting the continuous trajectories of corner points to guide the temporal registration of high-dynamic-range RGB frames. Registration plays a central role in compensating the intensity changes. Experimentally, the proposed event-assisted robust object tracking can work quite well in a high-dynamic-range environment that goes beyond the capability of RGB cameras.

The performance of the proposed algorithm is superior to both the counterpart taking only the RGB frames as input and that directly taking two inputs, and the advantages

hold in a wide range of applications. From the comparison, we can obtain the following two conclusions: (i) Under harsh illumination, the quality of RGB images greatly affects performance in downstream tasks. In order to ensure the robustness of performance in tasks such as object tracking, the RGB frames need to be enhanced first. (ii) Event signals, as lightweight and efficient sensors, can be used to capture critical information in high-speed-moving scenes. In addition, event signals are insensitive to lighting conditions and can be used for scene sensing under extreme illumination.

**Limitations.** The proposed algorithm mainly has two limitations. First, because of the involved complex calculations, it is difficult to deploy the algorithm into a UAV due to the limited arithmetic power. To achieve UAV deployment, it is necessary to further optimize the network structure for lightweight computation. Second, since the event camera can only capture the intensity changes in the scene, it is difficult to sense the targets being relatively stationary with respect to the event camera. Therefore, other complementary sensors need to be equipped for highly robust object tracking.

**Potential extensions.** In the future, we will dig deeper into the characteristics of event signals and construct neural networks that are more compatible with event signals to realize lightweight network design and efficient learning. In addition, we will integrate sensing units such as LIDAR and IMUs to achieve depth-aware 3D representation of scenes.

**Author Contributions:** Y.H. and J.S. conceived this project. Y.H. designed the framework and the network architecture. Y.H. and X.Y. implemented the event-based key feature alignment as well as temporal RGB image enhancement. H.L. collected the dataset and conducted the comparison experiments. X.Y. designed and conducted the ablation studies and analyzed the experimental results. J.S. dominated the discussion of this work. J.S. supervised this research and finally approved the version to be submitted. All authors have read and agreed to the published version of the manuscript.

**Funding:** This work was supported by Ministry of Science and Technology of China (Grant No. 2020AAA0108202), National Natural Science Foundation of China (grant number 61931012, 62171258).

**Data Availability Statement:** The data are available from the corresponding author upon reasonable request.

**Conflicts of Interest:** Author Heng Luan was employed by the company Research and Development Center, TravelSky Technology Ltd. The remaining authors declare that the research was conducted in the absence of any commercial or financial relationships that could be construed as a potential conflict of interest.

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
