# Peer review of "Event-Assisted Object Tracking on High-Speed Drones in Harsh Illumination Environment"

_drones, doi:10.3390/drones8010022_

Round 1

Reviewer 1 Report

Comments and Suggestions for Authors

The paper proposes an event-assisted robust object tracking algorithm which works in high-dynamic-range scenes. It integrates information from an event camera and an RGB camera to overcome the negative impact of harsh illumination on tracking performance and can achieve reliable tracking under large intensity variations.

The algorithm demonstrates superior performance compared to other algorithms, especially under harsh illumination conditions. It can track the object accurately even when the light is extremely weak. It can be a reliable choice for object tracking in challenging scenarios.

The paper is well written and has enough information to explain the problem and the solution. Although, it can better explore the algorithm limitations.

I only have minor suggestions:

One paragraph at the end of the introduction detailing the paper structure.

And to separate the experimental settings (3.1) of the section results (3.) as an another section "Methodology" (or something similar).

Add along the text the main limitations of the algorithm (a well defined scope). It can be linked with future work in the conclusion. 

Author Response

The authors would like to thank the reviewer for providing us with very positive evaluations regarding the manuscript entitled ``Event-Assisted Object Tracking on High-Speed Drones under Harsh Illumination Environment". We have revised the manuscript according to the comments and suggestions carefully, with the point-by-point responses listed below.

We have revised the manuscript according to the suggestions, including i) adding a paragraph at the end of the introduction to outline the paper's organization; ii) separating the “results” in Sec. 3.1 into a new section ("Experiment Settings"); iii) adding a paragraph at the end of the manuscript to discuss the main limitations of the algorithm.

All the revised parts are highlighted in red color for the reviewer to conveniently check. We hope the response letter solves all your concerns.

Reviewer 2 Report

Comments and Suggestions for Authors

The main objective of the presented work was to demonstrate a dual-sensing architecture that leverages the advantages of event cameras to raise the imaging quality of the RGB sensor as well as the successive object tracking performance, to extend the applicability of visible-wavelength cameras in real scenes. The presented results are interesting; however the revised manuscript needs to answer the following remarks:

1) The motivation behind the proposed work and the developed method for image processing should be precisely described and substantiated in the context of application in drones.

2) Each the contribution listed in bullet point in page 3 should precisely referenced to the state of the art, with indicated measurable key performance indicators, proving the progress beyond the SoA and the claimed advantages.

3) More explanation and details should be provided for Figure 4; in particular, what is the meaning of the abbreviations presented in the frame (conv, BN, ReLU, max pool, concat, up-conv).

4) The algorithm for the proposed method is presented only generally on block diagrams. A mathematical basement of the proposed algorithm should also be shown in the article.

5) It should be clearly and convincingly substantiated in the manuscript why these three baseline algorithms: Siamrpn++, RT-MDNet, and VisEvent, were chosen as benchmark for tracking performance comparison.

6) What is meant by saying “optimal” in line 283: “According to Tab. 1, the proposed algorithm demonstrates the optimal tracking results” ?

7) The quantitative performance of different object tracking algorithms in terms of PP and SP was presented only on the simulated dataset (Table 1). In the revised version of the manuscript, the advantage of the proposed method should be proved on more data, including the real data.

Reviewer 3 Report

Comments and Suggestions for Authors

The structure of the paper is very well established, going through all the sections that a proper research paper should have: introduction (literature review included here), methodology (Framework and Algorithm Design), experimental results (and associated discussions) and finally the conclusions.

The introduction contains all the relevant information about what the paper or the authors address throughout the manuscript. It is very well written, the authors analyzed sufficient relevant publications in the field of the paper, and they very cleverly highlighted the shortcomings of currently employed methods.

The methodology has a high scientific value, clearly describing the methods / techniques that were employed by the authors for their preset objective(s).

The results are very well presented from four standpoints: experiment settings, simulated data, real-world data and ablation studies. In the case of the second one, both qualitative and quantitative results can be found and are compared to well-established tracking algorithms. An improvement can be observed in this case and also in the first and third case (Fig. 8; Table 1; Figure 9).

The conclusions put forth by the authors are supported by the results.

However, apart from the above aspects, I can recommend the authors to further improve their manuscript by considering the following shortcomings:

- I recommend that the manuscript should undergo a thorough English language grammar and spellcheck correction, there are a few minor mistakes in this sense;

- the font of the paper should be consistent throughout the manuscript, currently the font in the figures is different from the main body of the text.

I congratulate the authors for their work.

Comments on the Quality of English Language

I recommend that the manuscript should undergo a thorough English language grammar and spellcheck correction, there are a few minor mistakes in this sense;

Author Response

The authors thank the reviewer for providing positive evaluations regarding the manuscript entitled ``Event-Assisted Object Tracking on High-Speed Drones under Harsh Illumination Environment". Following the reviewer's suggestion, we have modified the font in the figure to keep consistent with the main text and improved the writing carefully for a better presentation.

Round 2

Reviewer 2 Report

Comments and Suggestions for Authors

In the revised version of the article the Authors took into account the recommended amendments, thus making the work sufficiently suitable for publication in Drones.